# Polyphosphate discriminates protein conformational ensembles more efficiently than DNA promoting diverse assembly and maturation behaviors

**Saloni Goyal, Divya Rajendran, Anup Kumar Mani, Athi N Naganathan***

Department of Biotechnology, Bhupat and Jyoti Mehta School of Biosciences, Indian Institute of Technology Madras, Chennai, India

## eLife Assessment

This manuscript offers **important** insights into how polyphosphate (polyP) influences protein phase separation differently from DNA. The authors present **compelling** evidence that polyP distinguishes among protein conformational ensembles, leading to divergent condensate maturation behaviors that include unfolding and polyproline II formation. In response to reviewer feedback, the authors addressed key concerns by incorporating charge-equivalent DNA controls and extending structural analysis to FruR variants, further reinforcing the polymer-specific effects of polyP. While some discrepancies between protein systems remain unresolved, the study enhances our understanding of how biopolymers influence protein assembly and conformational transitions.

*For correspondence:
athi@iitm.ac.in

**Abstract** Disordered proteins and domains often assemble into condensates with polyanionic nucleic acids, primarily via charge complementarity, regulating numerous cellular functions. However, the assembly mechanisms associated with the other abundant and ubiquitous, anionic, stress–response regulating polymer, polyphosphate (polyP), are less understood. Here, we employ the intrinsically disordered DNA-binding domain (DBD) of cytidine repressor (CytR) from *E. coli* to study the nature of assembly processes with polyP and DNA. CytR forms metastable liquid-like condensates with polyP and DNA, while undergoing liquid-to-solid transition in the former and dissolving in the latter. On mutationally engineering the ensemble to exhibit more or less structure and dimensions than the WT, the assembly process with polyP is directed to either condensates with partial time-dependent dissolution or spontaneous aggregation, respectively. On the other hand, the CytR variants form *only* liquid-like but metastable droplets with DNA which dissolve within a few hours. Polyphosphate induces large secondary-structure changes, with two of the mutants adopting polyproline II-like structures within droplets, while DNA has only minimal structural effects. Our findings reveal how polyphosphate can more efficiently discern conformational heterogeneity in the starting protein ensemble, its structure, and compactness, with broad implications in assembly mechanisms involving polyP and stress response in bacterial systems.

## Introduction

Biomolecular condensates are mesoscale membrane-less structures that facilitate various cellular processes from stress response and signal transduction to RNA processing and chromatin organization (*Brangwynne et al., 2009*; *Hyman and Brangwynne, 2011*; *Shin and Brangwynne, 2017*; *Banani et al., 2017*; *Alberti and Hyman, 2021*). Physically, condensates are formed by the demixing or phase

**Figure 1.** Conformational features of CytR WT and mutants. (**A**) Molecular structure of polyP. (**B**) Amino acid sequence of CytR. Blue and red boxes indicate the positions of alanine and proline mutations in DM and P33A variants, respectively. (**C**) Disordered propensity of CytR predicted using IUPred3 (*Erdős et al., 2021*) with gray and black curves showing long and short disorder predictions, respectively. (**D**) Far-UV circular dichroism (CD) spectra of WT (green), DM (blue), and P33A (red) at 298 K in mean residue ellipticity (MRE) units of deg. cm² dmol⁻¹. (**E**) Thermal denaturation curves of CytR variants from far-UV CD experiments monitored at 222 nm and reported in MRE units. (**F**) Stokes radius following the color code in panel E. (**G**) Electrostatic potential map (*Jurrus et al., 2018*) of CytR in its folded conformation displaying a large positive electrostatic potential. (**H**) The hypothesis tested in the current work. WT (left), DM (middle), and P33A (right) ensembles could potentially form condensates or aggregates in the presence of polyP or DNA, and which could also display differential time-dependent properties. U, PF, and F represent unfolded, partially folded, and folded conformations, respectively.

separation of bio-macromolecules (i.e. proteins, DNA, RNA) into a polymer-rich or a condensed phase and a solvent-rich phase. The phase separation can be driven by homotypic or heterotypic interactions depending on the nature of demixing species, with a host of factors including the concentration and composition of constituent species, specific ions, ionic strength, temperature, and pH governing the transition (*Shapiro et al., 2021*; *Pappu et al., 2023*; *Zhou et al., 2024*). Both folded and disordered regions of proteins can form condensates, with a lot more propensity and prevalence noted in the latter. Biomolecular condensates are also implicated in numerous diseases, and specifically in neurodegenerative disorders, with the condensates or sometimes the dilute phase proposed to seed the formation of aggregates and amyloids (*Alberti and Hyman, 2021*; *Rai et al., 2021*; *Agarwal and Mukhopadhyay, 2022*; *Rangachari, 2023*; *Mukherjee et al., 2024*).

Complex coacervation, wherein oppositely charged macromolecules condense and promote phase transition, is often observed in DNA- and RNA-binding domains that harbor excess positive charges to bind the anionic counterpart (*Pappu et al., 2023*; *Larson et al., 2017*; *Strom et al., 2017*; *Vieregg et al., 2018*; *Dutagaci et al., 2021*; *King and Shakya, 2021*; *Phan et al., 2024*; *Nordenskiöld et al., 2024*). However, the role of inorganic polyphosphates (polyP), a series of phosphate units (Pi) linked together via phosphoanhydride bonds (*Figure 1A*), in condensate formation and maturation has been less explored. PolyP is a ubiquitous molecule that is produced under stress conditions like oxidative stress, heat shock, nutrition limitation, etc., in all organisms (*Kornberg et al., 1999*; *Brown and Kornberg, 2004*; *Rao et al., 2009*). In fact, the polymer length of polyP and the concentration vary among different species, ranging from a few tens to thousands, reaching up to 50 mM in bacterial cells under stress conditions (*Kornberg et al., 1999*). Apart from the stress response, it has been implicated in numerous processes ranging from modulation of nucleoid structure to biofilm formation (*Shi et al., 2004*; *Xie et al., 2019*; *Müller et al., 2019*; *Beaufay et al., 2020*; *Beaufay et al., 2021*; *Solesio et al., 2021*). Intriguingly, polyP is able to act both as a chaperone (*Gray et al., 2014*) (assisting folding of proteins, preventing protein misfolding and inhibiting aggregation) and in promoting aggregation (*Cremers et al., 2016*; *Yoo et al., 2018*), indicative of more complex underlying mechanisms (*Xie and Jakob, 2019*; *Guan and Jakob, 2024*) which could be protein specific.

Based on the observations above, we pose this question: Is polyphosphate more sensitive to the starting conformational ensemble than DNA, leading to differential behaviors of promoting aggregation versus acting as a chaperone? If so, and given the poly-anionic nature, does it lead to aggregates or condensates or both? To answer these questions, one needs a system that is also a DBD to enable direct comparison. Second, the DBD should be mutationally tunable to populate native ensembles with different structure, compactness, and helicity. Third, the charge composition and their sequence patterning should not be changed across the mutants, as this will naturally affect the binding to polyP. Here, we employ Cytidine repressor DBD (CytR DBD or CytR or CytR wildtype), an intrinsically disordered domain (*Figure 1B, C*) that adopts a folded conformation in the presence of DNA (*Moody et al., 2011*) as a model system to answer these questions. Despite its disordered nature, mutations of different types and at different positions can be introduced into CytR to make it adopt a continuum of conformations, from highly disordered to globally ordered (i.e. with a specific three-dimensional structure) (*Munshi et al., 2018c*; *Munshi et al., 2019*; *Munshi et al., 2020*). Specifically, the WT populates a minor excited state which is fully folded (~8–10%), but otherwise the native ensemble is disordered, with marginal helicity and a Stokes radius of 19.5 Å (green in *Figure 1D, F*; *Munshi et al., 2018c*; *Madhurima et al., 2023*). One of the mutants we consider is the double mutant of CytR (A29V/A48M; DM) which displays the properties of a fully folded compact domain driven by the enhanced hydrophobicity of the side chains at positions 29 and 48; these two mutations act synergistically to switch the ensemble from being disordered to fully ordered as shown earlier (*Munshi et al., 2019*). The DM therefore displays a higher helicity and lower Stokes radius of 17 Å, compared to the WT (blue in *Figure 1D, F*). The other mutant is P33A, in which the minor folded state population is eliminated because of the enhanced flexibility of alanine in the place of proline (*Munshi et al., 2018c*). The P33A mutant is nearly fully disordered as evidenced by the Stokes radius of 21.5 Å and minimum helicity (red in *Figure 1D, F*).

In addition, because CytR has an excess charge of +9 and a large positive electrostatic potential in its folded conformation (*Figure 1G*), it binds non-specifically to DNA (*Munshi et al., 2018b*; *Munshi et al., 2018a*). We therefore hypothesize that CytR should bind to any negatively charged polymer driven purely by electrostatic complementarity. In this work, we employ CytR WT and two mutants of CytR that exhibit diametrically opposing conformational preferences but identical charge composition and two anionic polymers (polyP and DNA) to understand the extent to which the starting ensemble (and hence the density of charges and stability or structure) dictates the nature of macromolecular assemblies and their maturation properties (*Figure 1H*). We find that the starting conformational ensemble is uniquely recognized by polyphosphate, leading to diverse outcomes including metastable condensates, aggregates, and time-dependent dissolution.

## Results

### CytR WT and DM form condensates with polyphosphate while P33A aggregates

Inorganic $polyP_{45}$, i.e. polyphosphate with 45 Pi units and with a net charge of –47, is employed in our studies and is referred to as polyP in the text below. We explored a range of solution conditions involving two-component mixtures of WT and polyP ([polyP] of 3–100 µM and [WT] of 5–125 µM at 20 mM phosphate buffer, pH 7 and 298 K) to map the phase diagram (*Figure 2A*). Solutions containing [WT] > 25 µM and [polyP] > 10 µM consistently showed higher turbidity (turbidity at 350 nm). We therefore chose [polyP] of 22 µM for further experiments. DIC (differential interference contrast) microscopy reveals droplet-shaped condensates with a diameter of ~1–4 µm on average (*Figure 2B*; also *vide infra*), which is also confirmed by fluorescence microscopy of NHS-rhodamine-labeled protein (*Figure 2C*). The number of droplets increases with increasing [WT] in the range between 25 and 90 µM consistent with OD measurements (*Figure 2D*). These observations establish that the WT CytR undergoes phase separation into condensates with polyP, and that turbidity can be employed as a reasonable proxy for phase separation in this system.

Given that CytR carries a large excess positive charge (+9) and polyP is an anionic polymer with no structural preference (unlike DNA), the interaction and the resultant assembly process is expected to be driven by non-specific electrostatic interactions. To test this, we use a [WT]:[polyP] of 90:22 µM and measure OD as a function of ionic strength at pH 7. The OD is observed to be independent

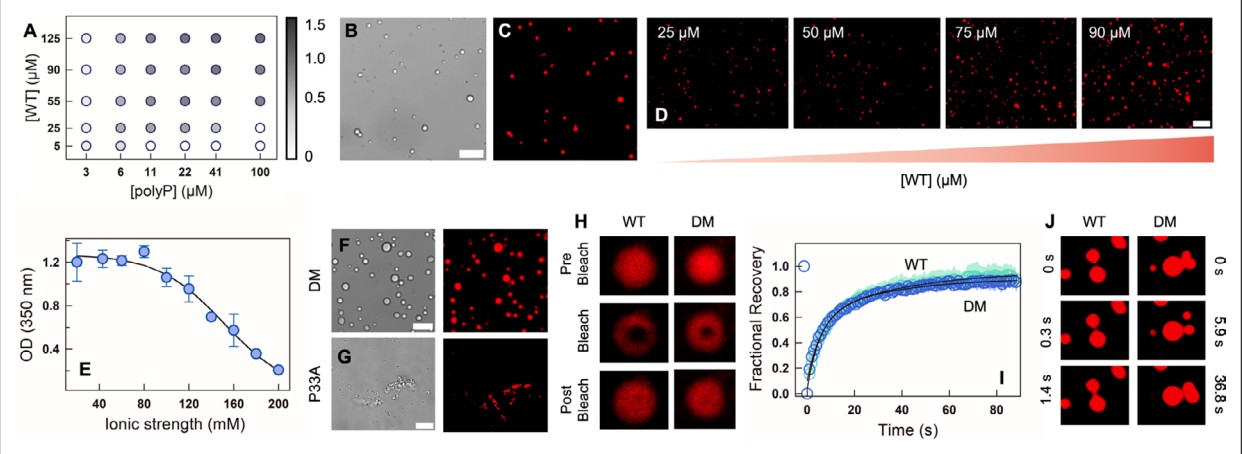

**Figure 2.** CytR undergoes phase separation with polyP in vitro. (**A**) Phase diagram illustrating the phase separation of WT at different protein and polyP concentrations. Empty circles represent no phase separation, and filled circles represent the extent of phase separation following the color bar which indicates the OD (turbidity) at 350 nm. Representative DIC (**B**) and fluorescence (**C**) microscopy images of WT showing condensate formation in the presence of polyP. (**D**) Fluorescence microscopy images of NHS-rhodamine labeled WT at different protein concentrations in the presence of 22 µM PolyP. The scale bar is 10 µm. (**E**) Ionic strength dependence of turbidity at fixed WT (90 µM) and polyP (22 µM) concentrations. The error bar indicates the spread from experimental replicates (N = 2). DIC (left) and fluorescence (right) microscopy images of NHS-rhodamine labeled DM (**F**) and P33A (**G**) in the presence of polyP. The scale bar is 10 µm. (**H**) Representative fluorescence microscopy images of condensates during fluorescence recovery after photobleaching (FRAP) on WT (left) and DM (right) immediately after polyP addition. (**I**) FRAP recovery curves at 0 hr for DM (blue) and WT (green). The data represents an average of five experiments (N = 5), and the errors (shaded areas) are smaller than the size of the circles. (**J**) Time-lapse fluorescence images showing a dripping event for the WT (left column) and fission–fusion events for DM (right column).

The online version of this article includes the following video and figure supplement(s) for figure 2:

**Figure supplement 1.** Fluorescence microscopy images of 90 µM WT with 22 µM polyP at 160 mM ionic strength (pH 7) immediately after adding polyP (left panel) and after 30 min (right panel).

**Figure 2—video 1.** Movie depicting droplet coalescence in condensates formed with CytR WT and polyphosphate, indicative of liquid-like properties. https://elifesciences.org/articles/105461/figures#fig2video1

**Figure 2—video 2.** Movie depicting droplet coalescence in condensates formed with CytR DM and polyphosphate, indicative of liquid-like properties. https://elifesciences.org/articles/105461/figures#fig2video2

of ionic strength until ~80 mM following which it linearly decreases with near-zero turbidity at 200 mM (*Figure 2E*). Under physiological ionic strength conditions of 150–160 mM, the OD value is still ~0.6 and with droplets evident in fluorescence microscopy (*Figure 2—figure supplement 1*).

The double mutant (DM), which is folded and compact, also undergoes spontaneous phase separation at the same concentrations as the WT when mixed with polyP, while the P33A mutant aggregates on addition of polyP (*Figure 2F, G*). The fluorescence of the NHS-rhodamine-labeled WT and DM recovers on photobleaching, highlighting the liquid-like nature of the condensates (*Figure 2H*). Specifically, the WT displays a maximal recovery of ~92% at 90 s (recovery half-time, $t_{1/2} = 6.0 \pm 2.5$ s), and the DM recovers to ~87% ($t_{1/2} = 5.2 \pm 1.3$ s) (*Figure 2I*). We also observe multiple coalescence and de-coalescence events in both the proteins at the earliest times (*Figure 2J*, *Figure 2—video 1* for WT and *Figure 2—video 2* for the DM).

Taken together, the fully unfolded variant of CytR (P33A) forms aggregates, and the variants with more 'folded-like' states undergo phase separation with polyP. These results indicate that charge composition and patterning alone do not determine the phase separation tendencies in this system. They underscore the importance of the starting ensemble, and potentially the population of partially structured substates, in determining the nature of the assembly process observed.

## Maturation of polyphosphate-induced assemblies

We carried out time-dependent studies in the presence of thioflavin T (ThT) enabling simultaneous measurement of OD and fluorescence in the same sample. The OD of WT–polyP mixtures shows a strong time dependence, with the OD decreasing from 1.2 to less than 0.9 in the first 10 min, but with little increase in ThT fluorescence (left panel in *Figure 3A*). However, the ThT fluorescence starts

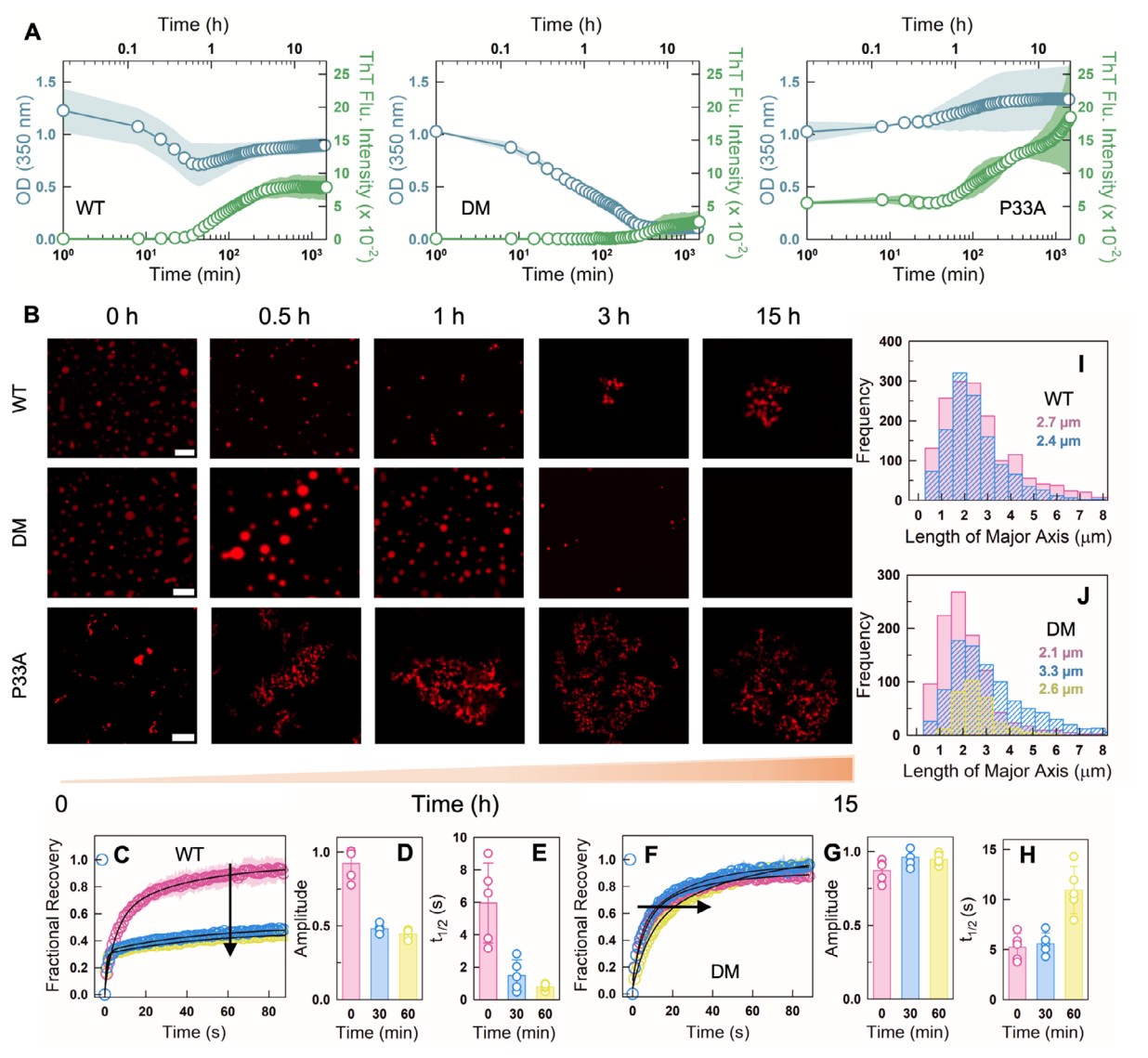

**Figure 3.** Time dependence of polyP-induced assemblies. (**A**) Turbidity (blue circles and left *y*-axis) and thioflavin T fluorescence intensity (green circles and right *y*-axis) time dependence for 90 µM WT (left panel), DM (middle panel), and P33A (right panel) in the presence of 22 µM polyP. The average from *N* = 2 experiments (circles) and the spread in the respective data (shaded region) are shown. (**B**) Representative fluorescence images of NHS-rhodamine labeled WT (top panel), DM (middle panel), and P33A (bottom panel) in the presence of polyP at different time points. The scale bar is 10 µm. Fluorescence recovery after photobleaching (FRAP) experiments for WT (panels **C**–**E**) and DM (panels **F**–**H**) in the presence of polyP at different time points – 0 min (pink), 30 min (blue), and 60 min (yellow) – and from five droplets (*N* = 5) and plotted as mean ± s.d. (**C**, **F**) The FRAP recovery curves of NHS-rhodamine labeled WT (panel **C**) and DM (panel **F**). The experimental errors (shaded areas) are also shown. (**D**, **G**) Recovery amplitudes or extents from FRAP experiments on the WT (panel **D**) and DM (panel **G**) at different time points. WT shows less recovery with time, indicating a liquid-to-solid transition, while the DM recovers fully. (**E**, **H**) FRAP recovery half-times for the WT (panel **E**) and DM (panel **H**) at the indicated time points. (**I**, **J**) Droplet size distribution of WT (panel **I**) and DM (panel **J**) at different time points following the same color code in panels **C**–**H**. The numbers within the plot represent the mean droplet dimensions at the corresponding time points.

The online version of this article includes the following figure supplement(s) for figure 3:

**Figure supplement 1.** Turbidity (blue circles and left *y*-axis) and thioflavin T (ThT) fluorescence intensity (green circles and right *y*-axis) curve for 90 µM WT (left panel), DM (middle panel), and P33A (right panel) in the presence of 5 µM ThT.

to increase in a sigmoidal fashion from 20 min. The OD, on the other hand, drops to a value of 0.75 at ~30 min following which it increases and stabilizes at a value of ~1. The WT–polyP mixture therefore forms condensates and matures into aggregates within 3 hr which is directly observable in microscopy images (*Figure 3B*), mirroring reports in numerous other systems that undergo droplet formation

followed by a liquid-to-solid phase transition (*Patel et al., 2015*; *Ray et al., 2020*; *Shen et al., 2023*; *Gracia et al., 2022*; *Dhakal et al., 2023*). Control experiments with protein alone did not reveal any changes in turbidity or aggregation in the entire 24-hr duration (*Figure 3—figure supplement 1*), pointing to a process entirely driven by polyP–protein interactions.

The OD of the DM–polyP mixture exhibits a similar decrease in OD starting from a value of ~1, but this trend extends beyond 3 hr, with the OD dropping to a value of ~0.1 at the longest time points (middle panel in *Figure 3A*). We do observe a marginal increase in ThT fluorescence after 2 hr, but the increase is not as significant as the WT. These observations point to a scenario wherein the DM forms condensates to start with, but which slowly dissolve with time (*Figure 3B*). On the other hand, the P33A mutant forms only aggregates as evidenced by the high OD and high ThT fluorescence across all time points in the 24-hr observation window (right panel in *Figure 3A, B*). Fluorescence recovery after photobleaching (FRAP) experiments follow the trends reported in OD-fluorescence measurements. In the WT, the recovery amplitude decreases from $0.92 \pm 0.11$ at the zeroth minute to just $0.43 \pm 0.04$ at 60 min (*Figure 3C, D*). The FRAP $t_{1/2}$, however, drops to shorter times at $t = 60$ min, indicating that there is a small subset of dynamic molecules (i.e. a small percentage of mobile fraction) contributing to the rapid recovery (*Figure 3E*). Alternately, the DM–polyP condensates recover near fully even at the first hour, with the recovery time slowing down by a factor of 2 (*Figure 3F–H*).

We further quantified the dimensions of the droplets from the microscopy images at three different time points: 0, 30, and 60 min. At $t = 0$ and for the WT–polyP condensates, the mean droplet dimensions are of the order of 2.7 μm, but with a heavy right tail spanning nearly 8 μm. This tail is diminished at $t = 30$ min, reducing the overall dimensions of droplets (~2.4 μm) and their number (smaller area under the curve in *Figure 3I*). The DM displays a behavior which is the exact opposite (*Figure 3J*): at $t = 0$ the mean dimensions of the condensates are 2.1 μm with no heavy right tail; larger droplets appear at $t = 30$ min contributing to the appearance of the heavy right tail and with the mean droplet size increasing to 3.3 μm. At longer times ($t = 60$ min), the number of droplets reduces, and so do their mean dimensions as a result of dissolution.

## Structural changes within the condensates span over 3 hr

The differential behavior of the three variants raises questions on the nature of structural changes upon condensate formation and maturation. The WT circular dichroism (CD) spectrum in the dilute phase (i.e. in the absence of polyP) is characterized by minima at 202 and 222 nm with ~17% of residual helical content (*Figure 1D*). Upon formation of condensates, the spectral shape shifts substantially to one with a minimum at 230 nm and with no spectral features at wavelengths <215 nm (*Figure 4A*), demonstrating that the WT adopts a non-helical and potentially an unfolded conformation in the condensed phase at initial times. The spectrum of DM is more representative of a folded protein with minima at both 208 and 222 nm (*Figure 1D*), and with the signal at 222 nm indicative of a protein with nearly 30% helical content. On mixing with polyP, the spectral shift resembles that of the WT (*Figure 4B*), again indicative of a structural change in the condensed phase. The P33A mutant, which is more unfolded (helicity ~10%), however, exhibits a more intense spectrum with a minimum at 225 nm which is representative of β-sheet-like structures (*Figure 4C*; see below). These differences are more evident when the signal at 222 nm is plotted as a function of time: the WT and P33A mutant exhibit a trend quite different from that of the DM, with the latter acquiring structure with time despite unfolding on adding polyP (*Figure 4D*).

Since the spectra measured with polyP are an effective average of structural signatures from both the condensed- and dilute phases and of different secondary-structure types, the spectral shapes cannot be directly interpreted. To extract structural signatures and the associated trends, we perform a global singular value decomposition of the spectral time series of the three variants. The first two components contribute 92% of the overall signal change (*Figure 4E*). Of this, the first component is the spectrum of an α-helix with clear minimum at 222 nm and another at 208 nm, and accounts for 71% of the signal change. The amplitude of this component decreases with time in the WT and P33A variant, implying that helical conformations are less favored in their assemblies; this feature also signals the population of an alternate structural form which is observable in the second component (*Figure 4F*). In the DM, the amplitude of the helical component increases with time as expected when the droplet dissolves due to preferential transport into the dilute phase. However, the helical content

recovers to just ~15% at 4 hr (*Figure 4F*), conceivably due to non-specific interaction with polyP resulting in partially structured states or due to a small fraction of oligomers.

The second component, accounting for 21% of the signal change, bears the features of polyproline II structures in unfolded conformations (*Shi et al., 2002*) with a maximum at 231 nm and a minimum at 207 nm (*Figure 4E*). There is a marginal increase in the amplitude of this component in the WT in the first hour, following which it stabilizes (*Figure 4G*). The amplitude of this component increases in the DM continuously even at 4 hr indicating a tussle between the helical and polyproline II-like structures (*Figure 4G*). Since the droplets are fully dissolved at 4 hr and by which time the ThT fluorescence starts to marginally increase, it is likely that smaller-sized aggregates (as the contribution of this component is only ~20% of the overall change) coexist with folded and partially structured helical conformations in the DM–polyP mixtures at longer times. In the P33A variant, the amplitude of the second component has a negative sign, and when multiplied with the second component (required for reconstructing the original spectra), it points to a strong contribution from β-turn or polyproline I conformations, mixed with β-sheet (*Kelly and Price, 2000*). This is further confirmed by Fourier-transform infrared (FTIR) spectroscopy on P33A, wherein a strong absorbance band is evident in the wavenumbers 1610–1630 cm$^{-1}$, which is absent in both the WT and DM at $t = 0$ (*Figure 4H–J*). At $t = 60$ min, the relative intensity of this band increases in the WT and P33A, while no change in spectral shape is evident for the DM.

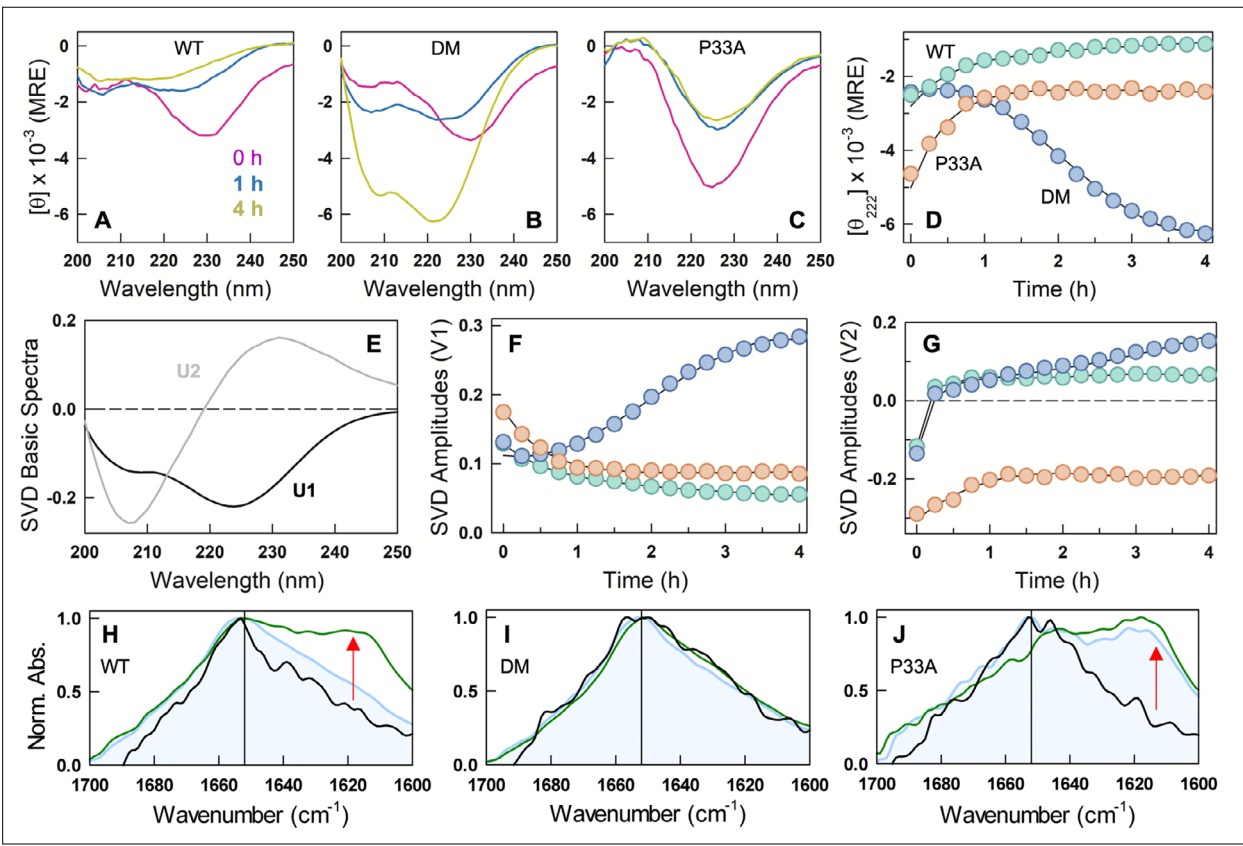

**Figure 4.** Structural changes in polyP-induced assemblies. (**A-C**) Far-UV circular dichroism (CD) spectra of 55 µM WT (**A**), DM (**B**), and P33A (**C**) at different time points – 0 hr (pink), 1 hr (blue), and 4 hr (yellow) – after the addition of 22 µM polyP at 298 K and displayed in mean residue ellipticity (MRE) units of deg. cm$^2$ dmol$^{-1}$. (**D**) Time dependence of the signal at 222 nm for the variants studied. Note the clear secondary-structure acquisition in the DM with time. (**E**) Basis spectra from a global singular value decomposition (SVD) of time-wavelength far-UV CD data. The first and second components are shown in black and gray, respectively. Amplitudes of first (**F**) and second (**G**) spectra as a function of time for CytR variants following the color code in panel **D**. Fourier-transform infrared (FTIR) spectra for WT (**H**), DM (**I**), and P33A (**J**) showing normalized absorbance recorded at the wavenumber range of 1700–1600 cm$^{-1}$. Black curves represent the FTIR spectra of protein in the absence of polyP. The blue and green curves are the FTIR spectra of protein after the addition of polyP at 0 and 60 min, respectively. Red arrows indicate the change in peak intensity after the addition of polyP, showing beta-sheet-like conformation in WT and P33A between 1610 and 1630 cm$^{-1}$. The vertical black line is at 1652 cm$^{-1}$, the amide I frequency indicative of helical structure.

In summary, we find that both the WT and the DM 'unfold' on forming condensates with polyP, with polyproline II-rich structures observed at longer times (~1–2 hr) in both the mixtures. The condensates thus formed are metastable, either undergoing a liquid-to-solid transition as in the WT or partially dissolving with time as in the DM. At the 4-hr time point, however, the DM acquires helical character due to the partial dissolution of the condensates, while the WT forms only aggregates. The starting ensemble thus determines the extent of metastability and the maturation of condensates, and not just the nature of assemblies formed.

## DNA is insensitive to the starting conformational ensembles

DNA is the primary anionic polymer bound by CytR, albeit without sequence preference and involving a distribution of binding free energies (*Munshi et al., 2018b*). We find that CytR WT also forms condensates with just 1 μM of double-stranded DNA in a concentration-dependent manner at 20 mM phosphate buffer, pH 7 (*Figure 5A, B*). The droplets are liquid-like, displaying coalescence behavior (*Figure 5C*), recovering fully on photobleaching in a few seconds and with an FRAP $t_{1/2}$ of 5.5±0.7 s (green in *Figure 5D, E*), comparable to the droplets formed in the presence of polyP. Both the DM and the P33A variants form phase-separated condensates with DNA (*Figure 5F, G*), unlike aggregated assemblies observed with polyP-P33A droplets (*Figure 2G*).

The time-dependent properties of the condensates with DNA again differ from those of polyP: the condensates dissolve within 90 min in both the WT and DM (*Figure 5H, I*), while taking nearly 3 hr for the P33A variant (*Figure 5—figure supplement 1*). Numerous droplets with a mean size of 1.6 μm form within the first few minutes (*Figure 5J*), which coalesce to form larger droplets of size 2–3 μm, following which they dissolve. The FRAP $t_{1/2}$ increases by a factor of two at 60 min compared to the initial time point (*Figure 5K*), and a very slow phase is also observed at 60 min (*Figure 5—figure supplement 2*), indicating ~20% immobile fraction. We find that OD is less sensitive to droplet formation with DNA, as we do observe few droplets even at 3 hr, a time point at which the OD is effectively zero (*Figure 5—figure supplement 1*). Both the DM and P33A mutants recover fully (amplitudes ~1) and with similar half-times compared to the WT at 60 min (*Figure 5—figure supplement 2*). To rule out the possibility of concentration differences (22 μM for polyP and 1 μM for DNA) contributing to the observed diverse behavior across the two anionic polymers, we carried out a set of control experiments at 11.2 μM DNA, a concentration at which the number of negative charges is equivalent to that of 22 μM polyP. However, the trends were identical to those observed at lower DNA concentration (*Figure 5—figure supplement 3*), further underscoring that the differences in chemical and conformational properties of the anionic polymers contribute to the diverse assembly behaviors.

None of the three proteins 'unfold' in the DNA-driven condensates; there is a marginal reduction (i.e. the signal becomes less negative) in the far-UV CD signal at $t = 0$ following which it quickly recovers to reach a value near that of the CD signal in the absence of DNA (*Figure 5L, M*). The absence of large structural change or aging into an aggregate with time is also evident in FTIR spectra which do not change significantly (*Figure 5—figure supplement 4*). The WT, however, does display 50% less negative signal at ~202–204 nm which shifts to longer wavelength after the first hour (blue and green spectra in *Figure 5L*), indicating helical structure acquisition and mirroring earlier reports (*Moody et al., 2011*; *Munshi et al., 2018c*). At higher DNA concentrations as used in the current study, this structural shift appears to be offset by unfolding in other regions likely through non-specific interactions. This leads to little signal change at 222 nm – note that this shift is not observable in the two mutants – and thus requiring detailed atomic-level studies to discern the subtle structural modulations induced by DNA at longer times.

To further test the lack of sensitivity of DNA to the starting conformational ensemble, we carried out a similar time-dependent experiment with polyP and DNA, but with the folded DBD of FruR (fructose repressor; *Figure 6A*) and the molten-globular variant Y19A FruR (*Rajendran et al., 2024*). PolyP spontaneously induces aggregation in FruR, while the molten-globular variant forms condensates but only at the earliest times (*Figure 6B–D*). The intensity of ThT fluorescence protein–polyP mixture increases with time, indicating that polyP induces a time-dependent enhanced aggregation of FruR, while the FruR Y19A–polyP mixture undergoes a rapid liquid-to-solid transition within the first 20 min (*Figure 6—figure supplement 1*). As with the CytR WT, the change in turbidity on aggregation happens earlier than the increase in ThT fluorescence (*Figure 6C*). The far-UV CD spectral patterns are different in the presence and absence of polyP, with the WT displaying altered secondary-structure

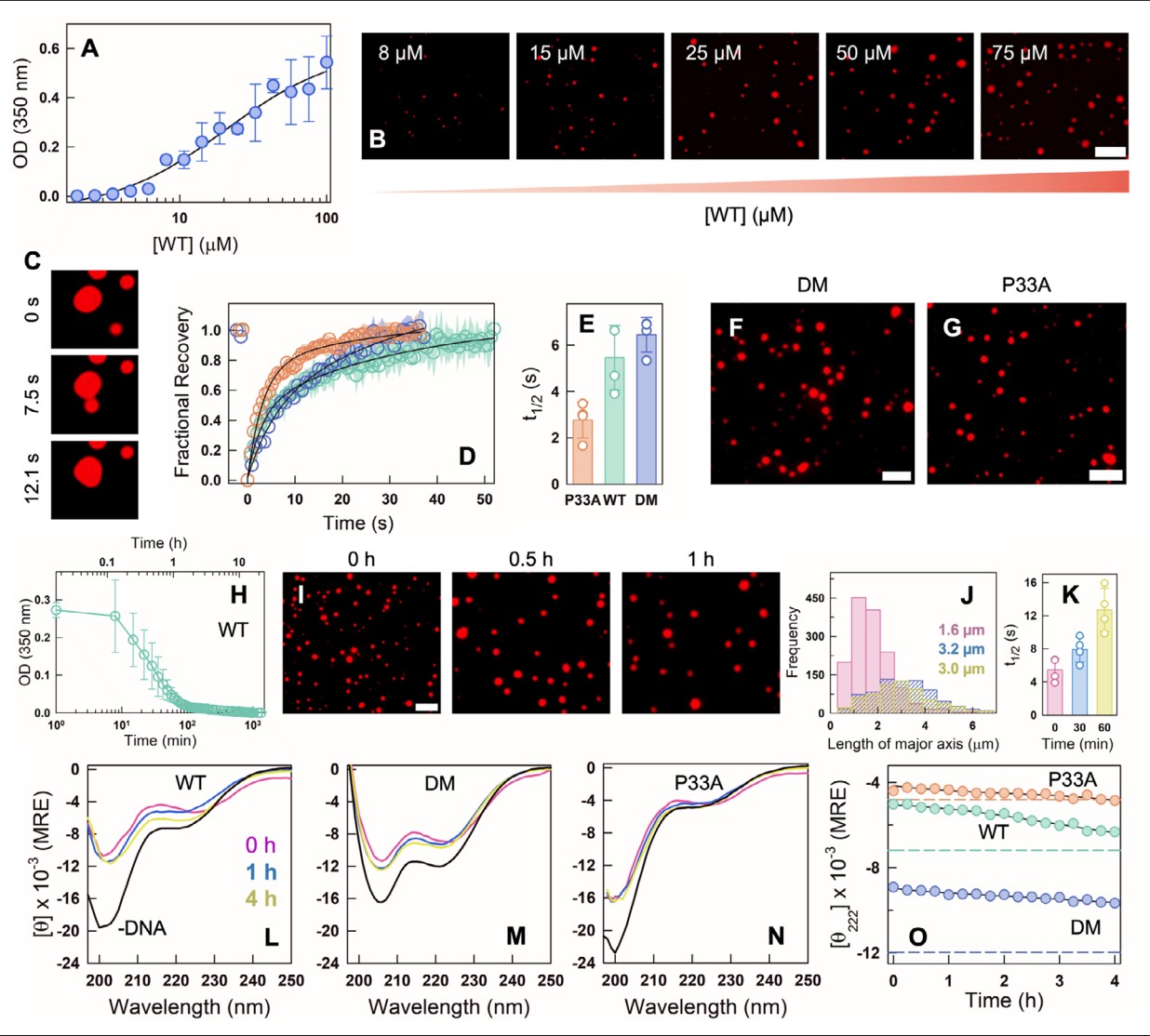

**Figure 5.** DNA induces metastable condensates that dissolve with time. (**A**) Changes in turbidity in solutions containing increasing concentrations of the WT and a fixed 1 μM concentration 45 bp specific DNA. The data is the mean from $N = 2$ experiments, and the error bar represents the spread. (**B**) Fluorescence microscopy images of NHS-rhodamine labeled WT at different concentrations with 1 μM DNA. The scale bar is 10 μm. (**C**) Representative time-lapse images of WT showing a fusion event. (**D**) Fluorescence recovery after photobleaching (FRAP) recovery curves of NHS-rhodamine-labeled CytR WT (green), DM (blue), and P33A (red). The data represents the average from $N = 4$ experiments. The experimental errors are shown as shaded areas. (**E**) Recovery half-times for the CytR variants at the earliest time points (0 min). P33A shows the fastest recovery, followed by WT and DM. Fluorescence microscopy images of 25 μM DM (**F**) and P33A (**G**) in the presence of 1 μM 45 bp DNA. The scale bar is 10 μm. (**H**) Turbidity of 25 μM WT with 1 μM DNA as a function of time. Data (circles) are from $N = 2$ experiments, and the error bar indicates the spread. (**I**) Fluorescence microscopy images of NHS-rhodamine labeled WT with DNA at different time points. The scale bar is 10 μm. (**J**) Droplet size distribution curve for WT in the presence of DNA at different time points – 0 min (pink), 30 min (blue), and 60 min (yellow). Numbers within the plot represent the mean droplet dimensions at the corresponding time points. (**K**) Recovery half-times from FRAP experiment on WT with DNA at different time points for $N = 4$ experiments following the color code for panel **J**. (**L–N**) Far-UV CD spectra of 25 μM WT (left), DM (middle), and P33A (right) at different time points – 0 hr (pink), 1 hr (blue), and 4 hr (yellow) – following the addition of 1 μM 45 bp DNA at 298 K. Black curve is the protein spectra in the absence of DNA. The data is reported in mean residue ellipticity (MRE) units of deg. cm² dmol⁻¹. (**O**) Time-dependent changes in far-UV CD signal at 222 nm for the CytR WT (green), DM (blue), and P33A (red). Dashed lines show MRE signal at 222 nm for the corresponding proteins in the absence of DNA.

The online version of this article includes the following figure supplement(s) for figure 5:

**Figure supplement 1.** Changes in OD and droplet dimensions as a function of time for DM and P33A variants, together with the fluorescence microscopy images of the three proteins in the presence of DNA at selected time points.

**Figure supplement 2.** Fluorescence recovery after photobleaching (FRAP) experiments for CytR variants in the presence of DNA at different time points: 0 min (pink), 30 min (blue), and 60 min (yellow).

*Figure 5 continued on next page*

*Figure 5 continued*

**Figure supplement 3.** Fluorescence microscopy images of 90 µM NHS-rhodamine labeled WT (top), DM (middle), and P33A (bottom) in the presence of 11.24 µM of 45 bp DNA at the time points indicated.

**Figure supplement 4.** Fourier-transform infrared (FTIR) spectra of WT (left), DM (middle), and P33A (right) showing normalized absorbance recorded at a wavenumber range of 1700–1600 cm$^{-1}$.

signature in the presence of polyP (*Figure 6E, F*). The Y19A mutant shows a significantly weakened helical structure content signaling structural destabilization and hence unfolding within the condensate (*Figure 6G, H*). In the presence of DNA, however, both the folded and the molten-globular variants display no change in turbidity with time (*Figure 6I–L*).

## Discussion

Phase separation of disordered CytR with the anionic polyphosphate is quite robust to solution conditions. However, when the native ensemble is perturbed via mutations to result in more order (DM) or less order (P33A) relative to the WT, polyP induces either phase separation or aggregation, respectively (*Figure 2*). The time-dependent behavior is markedly diverse with liquid-to-solid transition, partial dissolution, and enhanced aggregation in CytR, DM, and P33A, respectively (*Figure 3*). The trends are consistent across the different experimental protocols employed, including turbidity/fluorescence experiments, fluorescence microscopy, far-UV CD, and FTIR (*Figures 2–4*). On the other hand, DNA forms only metastable condensates that dissolve with time, despite different degrees of structural order in the native ensemble of CytR variants (*Figure 5*). To further prove that polyP interacts differently with ensembles, we performed experiments on the molten-globular variant of FruR (Y19A mutant) and the folded wild-type protein; we see two different behaviors with the WT aggregating and the mutant undergoing a liquid-to-solid transition with polyP, while DNA does not induce either aggregates or droplets (*Figure 6*).

The driving force for the different assembly processes is expected to have a strong contribution from the strength of intermolecular charge–charge interactions, the relative entropic penalty for inducing a conformational change in polyP versus that in the protein, release of counterions, and restructuring of water. The balance between these different enthalpic-entropic terms results in condensates being a more kinetically accessible phase but which are inherently metastable (as in the CytR WT, DM, and FruR Y19A), as a strong time dependence is visibly observable even in the first few minutes. With time, they access the globally stable phase, which is either the partially dissolved condensate or the more aggregated phase, conditions at which the chemical potential of the different constituent molecules equalize across the phases. Detailed molecular simulations that capture the relative experimental trends observed across the mutants are required to extract the contributions of each of the energetic and entropic terms.

Our observations establish that polyP is more sensitive to the conformational features of proteins than DNA, thereby contributing to the diverse outcomes. PolyP is expected to be highly flexible owing to its relatively simple chemical structure with only phosphoanhydride bonds that link adjacent units which are highly polar, while DNA has bulkier sugar and the apolar bases that limit its flexibility. It is therefore possible that polyP, owing to its flexibility, is able to wrap around the protein surface or even induce structural changes in proteins in a distinct manner depending on the nature of the starting ensemble. Strong evidence for the latter comes from our far-UV CD studies, wherein we find that the helical signature of the CytR variants is fully lost on adding polyP (*Figure 4*). Our results are consistent with previous observations of structural changes and 'unfolding' of proteins within condensates (*Murthy et al., 2019*; *Wen et al., 2021*; *Ubbiali et al., 2022*; *Joshi et al., 2023*). We additionally show that polyproline-II-like conformations are more probable within condensates formed by the CytR WT and the DM (*Figure 4*). One other observation stands out – the ThT fluorescence increases upon aggregation at a later time point than the changes in OD in CytR WT/DM and FruR Y19A; this is evidence that smaller aggregates are better captured by OD changes than ThT fluorescence, as the latter is more sensitive to larger aggregates.

The large charge density associated with polyphosphates and their natural abundance makes them prime candidates for inducing phase separation (*Wang et al., 2020*) through complex coacervation with proteins rich in basic amino acids. In fact, most DBDs which are typically associated

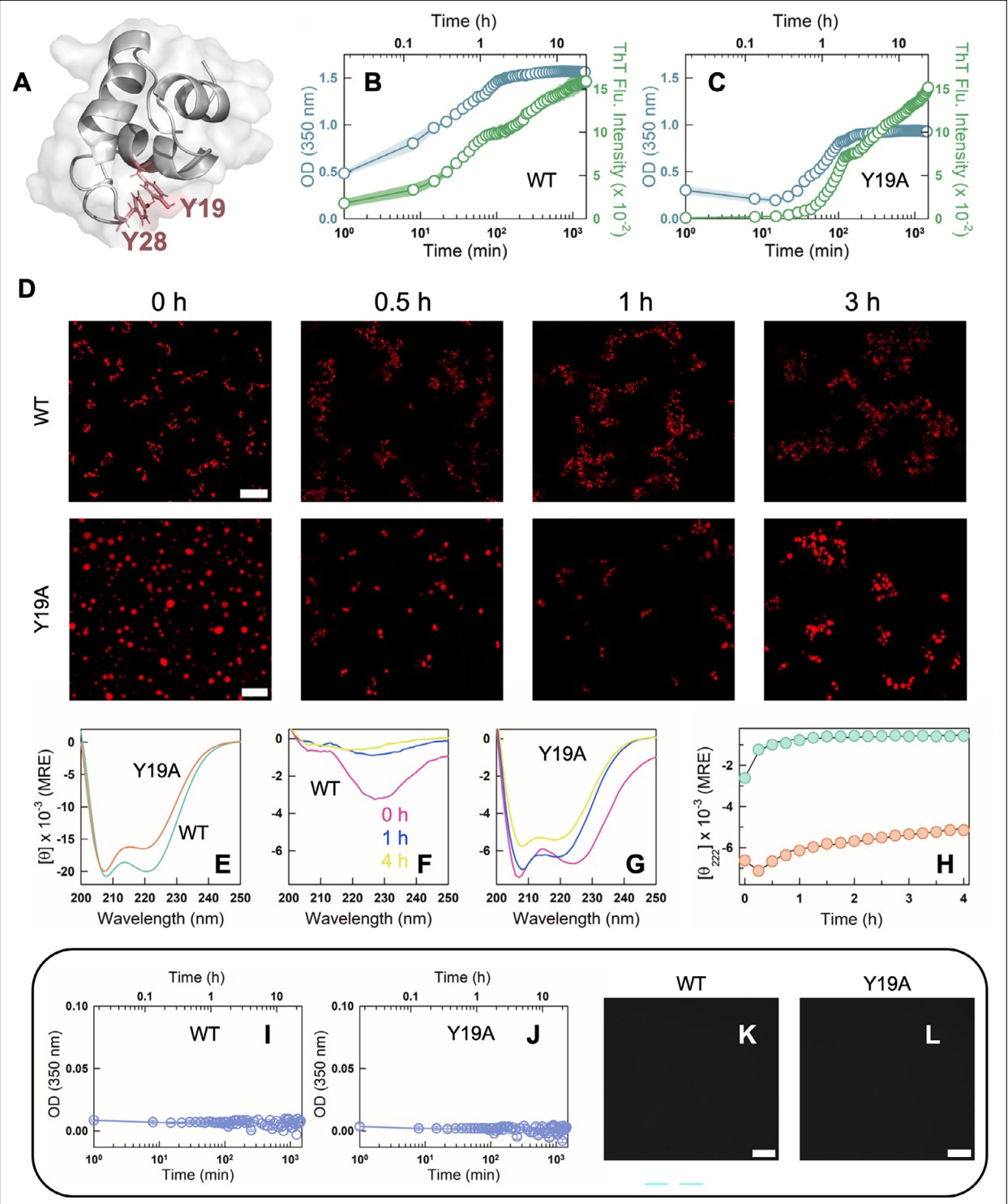

**Figure 6.** PolyP is able to discriminate between folded and molten-globular variants of FruR DNA-binding domain (DBD) while DNA does not. (**A**) 3D structure of the DBD of FruR highlighting an aromatic stacking interaction between Y19 and Y28. (**B,C**) Turbidity (blue circles and left *y*-axis) and thioflavin T (ThT) fluorescence intensity (green circles and right *y*-axis) curve for 90 µM FruR WT (**B**) and Y19A (**C**) in the presence of 22 µM polyP. The mean data from *N* = 2 experiments and the corresponding spread are shown as circles and shaded area, respectively. (**D**) Representative fluorescence microscopy images of NHS-rhodamine labeled FruR WT (top panel) and Y19A (bottom panel) in the presence of polyP at different time points (0, 0.5, 1, and 3 hr). The scale bar is 10 µm. (**E**) Far-UV circular dichroism (CD) spectra of FruR WT (green) and Y19A (red) at 298 K in mean residue ellipticity (MRE) units of deg. cm² dmol⁻¹ in the absence of polyP. Far-UV CD spectra of 55 µM FruR WT (**F**) and Y19A (**G**) at different time points – 0 hr (pink), 1 hr (blue), and 4 hr (yellow) – after the addition of 22 µM polyP at 298 K and displayed in mean residue ellipticity (MRE). Note that the *y*-axis scales of this panel

*Figure 6 continued on next page*

*Figure 6 continued*

are different from panel E. (**H**) Time dependence of the signal at 222 nm for WT (green) and Y19A (red). Turbidity of 25 µM WT (**I**) and Y19A (**J**) with 1 µM DNA as a function of time from *N* = 2 experiments. No changes in turbidity are observed with time. Representative fluorescence microscopy images of NHS-rhodamine labeled FruR WT (**K**) and Y19A (**L**) in the presence of DNA. The scale bar is 10 µm.

The online version of this article includes the following figure supplement(s) for figure 6:

**Figure supplement 1.** Fluorescence recovery after photobleaching (FRAP) experiments on the Y19A variant of FruR in the presence of polyP at different time points: 0–10 min (pink), 10–20 min (blue), and 20–30 min (yellow).

with transcription factors are rich in basic amino acids. This opens up a fascinating possibility where polyP and DNA could compete with transcription factors or work together forming ternary condensates regulating gene expression, as shown in studies on the bacterial protein Hfq (*Beaufay et al., 2021*). Taken together with our results on CytR, caution needs to be exercised in interpreting in-cell microscopy results, as it is possible that polyP is involved in condensates, especially when proteins of interest are rich in basic residues. Finally, the full-length CytR plays an important role in transcription regulation of genes involved in stress response, including critical roles in nucleoside catabolism, context-dependent activation or repression, and pathogenesis (*Barbier and Short, 1992*; *Rasmussen et al., 1996*; *Hirakawa et al., 2020*). CytR also binds to the promoter region of the rpoH gene and represses transcription, thus regulating the expression of σ (*Xie and Jakob, 2019*) – an RNA polymerase subunit essential for the transcription of heat shock proteins (*Kallipolitis and Valentin-Hansen, 1998*; *Lauritsen et al., 2021*). Given the results of the current work and since polyP concentrations rise significantly in stressed cells (*Kornberg et al., 1999*), it is tempting to speculate that polyP interacts with CytR to form condensates and sequesters it, thus promoting σ (*Xie and Jakob, 2019*) transcription. Studies on CytR combining other stress response regulators, specifically with the nucleoid-associated sensory protein H-NS, with DNA and polyP could unravel hitherto unexplored mechanisms regulating the nuanced gene expression patterns in enterobacteria.

# Materials and methods

## Key resources table

| Reagent type (species) or resource | Designation | Source or reference | Identifiers | Additional information |
|---|---|---|---|---|
| Sequence-based reagent | 45 bp long ds DNA – CytR specific | This paper | Oligonucleotide | 5' TGGTGGGTAAATTTATGCA ACGCATTTGCGTCAT GGTGATGAGTA 3' |
| Sequence-based reagent | 45 bp long ds DNA – FruR specific | This paper | Oligonucleotide | 5' TAAAGACAAGATCG CGCTGAAACGTTTCAA GAAAGCATAATACTT 3' |
| Recombinant protein | Glucose oxidase | Sigma-Aldrich | G2133-10KU, 10000units | From Aspergillus niger |
| Recombinant protein | Catalase | Sigma-Aldrich | E3289-100mg | From bovine liver |
| Chemical compound | Sodium phosphate glass – type 45 (polyphosphate) | Sigma-Aldrich | S4379-500MG | |
| Chemical compound | NHS-rhodamine | Thermo Fisher Scientific | 46406 | |
| Chemical compound | D-(+)-Glucose | Sigma-Aldrich | SLBR5156V | |
| Chemical compound | DL-Dithiothreitol | HiMedia | MB070-25G | |
| Chemical compound | Thioflavin T | Sigma-Aldrich | T3516-5G | |

## Purification of CytR DBD and mutants

The gene corresponding to the DBD of CytR(MKAKKQETAATMKDVALKAKVSTATVSRALMNPDKVS QATRNRVEKAAREVGYLPQPMGRNVKRNE) in the pTXB1 vector was transformed into *E. coli* BL21 (DE3) cells and purified as described before (*Munshi et al., 2018b*). Identical protocols were employed for purification of the DM and P33A variants. The Stokes radius was estimated as detailed in an earlier work (*Munshi et al., 2020*).

## Time-dependent scattering (OD) and ThT fluorescence measurements

Different concentrations of CytR WT were prepared by dissolving the lyophilized protein in 20 mM sodium phosphate, pH 7 (43 mM ionic strength) buffer. To generate the phase diagram, the scattering intensity (optical density) was recorded at 350 nm at protein concentrations of 5, 25, 55, 90, and 120 µM, and at different polyP concentrations of 3, 6, 11, 22, 41, and 100 µM. The measurements were acquired for 24 hr at 7-min intervals in a microplate reader (Agilent BioTek Synergy H1) at 298 K. The sample volume in each well was 200 µl. Further studies employed 90 µM protein (WT and mutants) and 22 µM polyP, unless mentioned otherwise. For the time-dependent studies recording both scattering and fluorescence, a 5-µM ThT (Sigma-Aldrich) was also added to the protein and polyP mixture; the wells were also excited at 442 nm, and the fluorescence intensity recorded at an emission wavelength of 485 nm. Ionic strength-dependent experiments were carried out at 20 mM sodium phosphate buffer, pH 7.0 (43 mM effective ionic strength) with the addition of NaCl for final ionic strength values spanning 60–200 mM. The OD values are reported directly from the plate reader without correction for path length, which is 5 mm in this experiment.

For the scattering measurements in the presence of DNA, a range of CytR WT concentrations (2–100 µM) was used, and measurements were acquired with 1 µM of 45 bp dsDNA (5′ TGGTGGGT AAATTTATGCAACGCATTTGCGTCATGGTGATGAGTA 3′). Further studies with DNA were done at 25 µM protein (WT and mutants) concentration.

## NHS-rhodamine labeling

NHS-rhodamine (Thermo Fisher Scientific, USA) was dissolved in DMSO, and protein was dissolved in conjugation buffer at pH 7.4 (as per the manufacturer's protocol). The protein solution was added to 5×–7× molar excess of dye and incubated at 25°C for 2 hr at mild shaking conditions in the dark. The excess dye was removed by desalting using a 26/10 HiPrep desalting column (Cytiva). The labeled protein was eluted in 20 mM sodium phosphate buffer at pH 7.

## Flow chamber preparation for imaging experiments

Imaging experiments were done using 22 × 22 mm coverslips and glass slides (Blue Star, India). Coverslips and glass slides were kept in Piranha solution (3:1 vol/vol sulfuric acid and hydrogen peroxide) for 30 min and then washed thoroughly with MilliQ water. The slides and coverslips were then air-dried and wiped with absolute ethanol. Clean slides and coverslips were used to prepare flow chambers to observe the condensate formation. To prepare the flow chamber, the double-stick tape was sandwiched between the glass slide and coverslip. The chamber was then sealed from three sides to avoid sample leakage.

## Fluorescence microscopy

The condensates were observed under 100×/1.45 NA oil immersion objective at room temperature using Olympus IX83 inverted fluorescence microscope. 10% of the labeled protein was mixed with unlabeled protein, and the sample was loaded into the flow chamber immediately after adding polyP/DNA. An anti-fading cocktail containing 20 µg/ml glucose oxidase, 1.4 µg/ml catalase, 20 mM glucose, and 10 mM dithiothreitol was used to prevent photobleaching of the sample (*Sudhakar et al., 2021*). The sample was imaged at different time points under DIC and fluorescence mode using an appropriate fluorescence channel at 2048 × 2048 pixels (images for WT in the presence of polyP were acquired at 1024 × 1024 pixels) with 8-bit depth resolution. Images were analyzed using cellSens Dimensions Desktop (provided with the instrument) and ImageJ (*Schneider et al., 2012*).

## Fluorescence recovery after photobleaching

For FRAP experiments, 10% and 5% of labeled protein were used for polyP and DNA studies, respectively. Anti-fading cocktail was added to the protein sample to prevent passive photobleaching. The samples were loaded into the flow chamber at different time points (0, 30, and 60 min), and the experiment was performed in an Olympus Fluoview FV3000 confocal microscope with a 100×/1.45 NA oil immersion objective. A 561-nm laser was used to visualize and bleach (with 100% laser power) the condensates at the center. Fluorescence intensity was acquired for five different regions of interest (ROIs) with the same diameter: ROI-1 for the actual bleaching region, ROI-2 for the neighboring condensate for passive bleaching correction, and ROI-3, 4, and 5 outside the condensates to record

background intensity for background correction. The images were acquired at 1024 × 1024 pixels with 12-bit resolution. Data was processed using cellSens Dimensions Desktop software and MATLAB. The raw data was corrected using the following equation (*Poudyal et al., 2023*):

$$I\left(n\right) = \frac{I\left(t\right) - I\left(b\right)}{I_c/I_{c_o}},$$

where $I\left(t\right)$ is the fluorescence intensity at time $t$, $I\left(b\right)$ is the average of background intensity for ROI-3, 4, and 5, $I_c$ is the fluorescence intensity of ROI-2 after photobleaching, and $I_{c_o}$ is the fluorescence intensity before photobleaching. The data was normalized ($I$) using:

$$I = \frac{I_n - I_{n_{min}}}{I_{n_o}},$$

where $I_n$ is corrected fluorescence intensity at time t and $I_{n_o}$ is the intensity before bleaching and $I_{n_{min}}$ is the minimum intensity. The average of corrected and normalized fluorescence intensity from different experiments was taken, and the data was plotted as a function of time.

## Circular dichroism

Far-UV CD spectra were recorded at protein concentrations of 55 and 25 µM immediately after the addition of 22 µM polyP and 1 µM DNA, respectively. The spectra were acquired as a function of time for 4 hr at an interval of 15 min in the wavelength range of 250–200 nm at 298 K with a scanning speed of 10 nm/min in a Jasco J-815 spectropolarimeter. The mean helical content for the different CytR variants is estimated by taking the ratio between the signal at 222 nm with that expected of a 100% helical protein (–40,000 MRE units of deg. cm$^2$ dmol$^{-1}$).

## FTIR spectroscopy

FTIR scans were performed using a Perkin Elmer Spectrum Two Spectrometer equipped with a DTGS detector in the attenuated total reflectance mode. 5 µl of protein sample was added to the diamond crystal, and the spectrum was acquired in the range of 4000–400 cm$^{-1}$ using an average of 50 scans at a resolution of 4 cm$^{-1}$. Baseline correction was done with MilliQ water to minimize the interference due to H$_2$O before spectral acquisition.

## Acknowledgements

The authors are grateful for the support from the Department of Biotechnology (DBT, India) for the grant BT/PR41973/BRB/10/1967/2021 to ANN. The authors acknowledge the FIST facility (SR/FST/LS-II/2020/552(C)) sponsored by the Department of Science and Technology (DST, India) and the ICSR – Common Instruments Facility (CIF) at the Department of Biotechnology, IIT Madras (Chennai, India) for the instrumentation. The authors thank Vijay Rangachari and Ethayaraja Mani for the discussions, Swathi Sudhakar for the help in flow chamber preparation, and Thalappil Pradeep and Sonali Seth for help in FTIR experiments. DR acknowledges Women Leading IITM for the funding.

## Additional information

### Funding

| Funder | Grant reference number | Author |
| --- | --- | --- |
| Department of Biotechnology, Ministry of Science and Technology, India | BT/PR41973/ BRB/10/1967/2021 | Athi N Naganathan |
| Department of Science and Technology, Ministry of Science and Technology, India | SR/FST/LS-II/2020/552(C) | Athi N Naganathan |

| Funder | Grant reference number | Author |
|--------|------------------------|--------|

The funders had no role in study design, data collection, and interpretation, or the decision to submit the work for publication.

## Author contributions

Saloni Goyal, Data curation, Software, Formal analysis, Validation, Investigation, Visualization, Methodology, Writing – original draft, Writing – review and editing; Divya Rajendran, Formal analysis, Investigation, Methodology; Anup Kumar Mani, Investigation; Athi N Naganathan, Conceptualization, Resources, Data curation, Software, Formal analysis, Supervision, Funding acquisition, Validation, Investigation, Visualization, Methodology, Writing – original draft, Project administration, Writing – review and editing

## Author ORCIDs

Athi N Naganathan ⬚ https://orcid.org/0000-0002-1655-7802

Reviewer #1 (Public review): https://doi.org/10.7554/eLife.105461.3.sa1
Reviewer #2 (Public review): https://doi.org/10.7554/eLife.105461.3.sa2
Author response https://doi.org/10.7554/eLife.105461.3.sa3

# Additional files

## Supplementary files

MDAR checklist

Source data 1. Source data for main text figures.

Source data 2. Source data for figure supplements.

## Data availability

All data generated or analyzed during this study are included in the manuscript and supporting files; source data files have been provided for all figures.

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
