## [Editor Report · eLife Assessment]

This manuscript offers **important** insights into how polyphosphate (polyP) influences protein phase separation differently from DNA. The authors present **compelling** evidence that polyP distinguishes among protein conformational ensembles, leading to divergent condensate maturation behaviors that include unfolding and polyproline II formation. In response to reviewer feedback, the authors addressed key concerns by incorporating charge-equivalent DNA controls and extending structural analysis to FruR variants, further reinforcing the polymer-specific effects of polyP. While some discrepancies between protein systems remain unresolved, the study enhances our understanding of how biopolymers influence protein assembly and conformational transitions.

---

## [Referee Report · Reviewer #1 (Public review)]

In the article Goyal and colleagues investigate the role of negatively charged biopolymers, i.e., polyphosphate (polyP) and DNA, play in phase separation of cytidine repressor (CytR) and fructose repressor (FruR). The authors find that both negative polymers drive the formation of metastable protein/polymer condensates. However, polyP-driven condensates form more gel- or solid-like structures over time while DNA-driven condensates tend to dissipate over time. The authors link this disparate condensate behavior to polyP-induced structures within the enzymes. Specifically, they observe the formation of polyproline II-like structures within two tested enzyme variants in the presence of polyP. Together, their results provide a unique insight into the physical and structural mechanism by which two unique negatively charged polymers can induce distinct phase transitions with the same protein. This study will be a welcomed addition to the condensate field and provide new molecular insights into how binding partner-induced structural changes within a given protein can affect the mesoscale behavior of condensates.

---

## [Referee Report · Reviewer #2 (Public review)]

Summary:

In the article Goyal et al. investigate how protein/polymer phase transition behavior is modulated by different binding partners-specifically, DNA and polyphosphate (PolyP). The authors show that while both DNA and PolyP can induce metastable condensates, only PolyP drives unique phase transition behaviors by effectively discriminating among initial protein ensembles with varying degrees of conformational heterogeneity, compactness, and plasticity. This selectivity is attributed to PolyP's ability to unfold the enzyme during condensate formation, supported by the observation of polyproline II-rich structures in two tested variants (CytR WT and DM). Overall, this work offers valuable insights into the mechanistic factors underlying condensation assembly and advances our understanding of how molecular interactions influence phase behavior.

Strengths:

The authors employed a well-designed and technically sound experimental approach to investigate how the initial protein conformational ensemble influences phase transition behavior in the presence of two charged polymers. Specifically, they examined phase transitions of CytR and FruR variants in the context of either polyphosphate (PolyP) or DNA, enabling a direct comparison that effectively highlights key differences. This study provides mechanistic insights into the role of PolyP in driving condensation and may contribute to a broader understanding of assembly processes involving PolyP, particularly in the context of bacterial stress responses.

Weaknesses:

The primary weakness of this manuscript lies in the lack of a consistent trend linking the unique phase transitions observed in protein/PolyP systems to the initial protein conformational ensemble. The observed differences in assembly and maturation behavior do not consistently correlate with conformational heterogeneity, plasticity, or compactness of the starting ensemble. This is particularly evident in the divergent outcomes between the CytR/PolyP and FruR/PolyP systems. Consequently, the phase behavior of protein/PolyP condensates does not reliably reflect the composition of the initial conformational ensemble, limiting its effectiveness as a probe for conformational state characterization.

---

## [Author Response]

The following is the authors’ response to the original reviews.

**Reviewer 1:**
In the article titled "Polyphosphate discriminates protein conformational ensembles more efficiently than DNA promoting diverse assembly and maturation behaviors," Goyal and colleagues investigate the role of negatively charged biopolymers, i.e., polyphosphate (polyP) and DNA, play in phase separation of cytidine repressor (CytR) and fructose repressor (FruR). The authors find that both negative polymers drive the formation of metastable protein/polymer condensates. However, polyPdriven condensates form more gel- or solid-like structures over time while DNA-driven condensates tend to dissipate over time. The authors link this disparate condensate behavior to polyP-induced structures within the enzymes. Specifically, they observe the formation of polyproline II-like structures within two tested enzyme variants in the presence of polyP. Together their results provide a unique insight into the physical and structural mechanism by which two unique negatively charged polymers can induce distinct phase transitions with the same protein. This study will be a welcomed addition to the condensate field and provide new molecular insights into how binding partner-induced structural changes within a given protein can affect the mesoscale behavior of condensates. The concerns outlined below are meant to strengthen the manuscript.Recommendation:

We value the reviewer’s positive comments and appreciate time taken to provide detailed feedback that has certainly helped improve our manuscript.

Major Concerns:(1) The biggest concern in this manuscript lies with experiments comparing polyP45, which has a net negative charge of -47, and double-stranded DNA of 45 base pairs (as stated in the methods), which will have a net negative charge of -90. Given the dependence of phase separation and phase transitions on not only net charge but charge density, this is an important factor to consider when comparing the effect of these molecules. It is unclear how or if the authors considered these factors in the design of their experiments. Because of the factor of 2 difference in net charge over the same number of polymer chain components, i.e. a chain of 45 pi vs. a chain of 45 double-stranded base pairs, it is unclear if the results from polyP vs. DNA are directly comparable. One solution would be to repeat all DNA experiments using single-stranded DNA so that the net charge is similar to polyP over the same chain length. Another possibility would be to repeat DNA experiments using a doublestranded DNA of 23 base pairs. This would allow for a nearly equal net charge (-46 vs. -47 for polyP), but the charge density would still be 2X polyP. As it stands now, the perceived differences in DNA vs. polyP behavior may be an artifact arising from the difference in net charge and charge density between DNA and polyP.

To address the reviewer’s concerns regarding charge density differences between polyP and DNA, we conducted an experiment using a higher DNA concentration (11.24 µM) to obtain charge equivalence between the two experiments (i.e. the total concentration of charges). As shown in Figure S5, even at higher DNA concentration, the condensates undergo progressive dissolution over time. This observation indicates that the differential maturation of condensates, arising from distinct initial protein ensembles, are governed by the intrinsic properties of polyP. Charge density (i.e. the number of charges per unit volume of the polymer), on the other hand, is an intrinsic feature of the polymer which is naturally different between DNA and polyP. In fact, the primary result of our work is our observation that polyP can discern the starting ensembles more efficiently, likely through actively engaging and interacting with the ensemble while DNA appears to be a passive player. The differences are not an artifact as they arise from fundamental features of two natural anionic polymers found within cells. In other words, the outcomes could be very different if the concentration of one polymer dominates over the other (see the response below).

(2) One outstanding question the authors do not address relates to how mixtures of CytR or FruR, DNA, and polyP behave. In the bacterial cytoplasm, these molecules are all in the same compartment (admittedly that compartment is not well mixed due to unique condensate-driven organization). Would the authors expect to see similar effects of polyP and DNA if they were in the same solution? Perhaps the authors could run a set of experiments where they vary the ratios of DNA and polyP to probe how increased levels of "stress", i.e. increased levels of polyP vs. DNA, alter the formation and behavior of enzymatic condensates.

Following this comment, we investigated the phase separation behavior of CytR WT in the presence of different charge ratios of polyP-DNA mixtures. As seen in Author response image 1,panel A below, the outcomes are highly sensitive to the starting concentrations: at higher charge concentration of polyP (left panel), the OD and ThT fluorescence intensity is high at lower time points, both decrease and increase again. Fluorescence microscopy images (panel B) reveal similar trends, but the more fascinating outcome are the FRAP recovery profiles which recover extremely fast and fully at zero time point (panel C) despite aggregation-like tendencies observed in ThT fluorescence assays. However, at longer time points (20 and 40 mins) the FRAP recovery is significantly weaker but recovers to ~65% at 1 hour (panel C). At high relative polyP concentrations with respect to DNA, droplets are formed first which then transition into aggregates (liquid-to-solid transition; middle image in panel A). At relatively high DNA concentrations it appears that both droplets and aggregates co-exist as both OD and ThT fluorescence are moderately high. Given these complex behaviors, we have not included the same in the current manuscript as we still do not fully understand the origins of these differences. In fact, we are planning to extend this study by exploring the combinations in detail to understand the relative roles played by the two polymers in ternary mixtures.

(3) In Figure 1H, the recovery trace shows the fractional recovery of DM to near WT levels. It is clear from the images that recovery of the bleached region occurs, but the overall fluorescence intensity of DM is much lower than WT, even when accounting for the difference in starting condensate sizes in the Pre-Bleach images. Shouldn't this qualitative difference in total fluorescence be reflected in the quantitative trace?

In Figure 2H, as the reviewer rightly points out, there is a clear difference in the absolute fluorescence intensity between WT and DM condensates. We would like to clarify that the recovery traces shown in Figure 2I were normalized to the pre-bleach intensity of each individual condensate to reflect fractional recovery. This normalization is intended to highlight the relative mobility of the protein within each condensate, but it does not capture the difference in total fluorescence intensity between WT and DM.

(4) A description of the molten-globular variant Y19A FruR should be included in the main text where the variant is introduced. There is currently no additional description of the molten-globular variant in the Supplement as suggested by the manuscript.

Figure 6A depicts the three-dimensional structure of FruR WT, with tyrosine residues Y19 and Y28, shown in red, forming stacking interactions. In the Y19A mutant, the loss of these interactions results in little changes in secondary structure (as shown in Figure 6E) but disrupts the protein’s tertiary structure, resulting in a molten globular state. The FruR work is now published in JPCB and can be found at https://doi.org/10.1021/acs.jpcb.4c03895, and is also appropriately cited in the revised version (reference 53).

(5) Throughout the manuscript, the authors discuss polyP and DNA being able (or unable) to "distinguish" between different variants of CytR and FruR. This is confusing and suggests that DNA or polyP can choose to bind one form over another. The authors should re-work the language in this section to better reflect their direct observations for the behavior of protein in CD experiments and condensate behavior in imaging and turbidity experiments.

We have now modified the text where necessary. The experiments were not done in the presence of both polyP and DNA, but in isolation (protein + polyP or protein + DNA). Hence, our aim is to convey that polyP is the polymer that leads to variable outcomes because of its ability to ‘interact’ differently with the different starting ensembles.

Minor Concerns:(1) For all Figures, please include the number of measurements, i.e., N = ...

We have updated all figure legends to include the number of measurements, indicated as *N* = ..., as suggested.

(2) For all Figures, please place panel labels, i.e., A, B, C, etc., in the same respective location for each panel. As currently mapped out, it is difficult to easily determine which data are associated with each panel because the IDs are in various locations.

Due to variations in data presentation and spacing within individual plots, it was challenging to place all labels in exactly the same position without obscuring important details. We have therefore maintained the labels as they were before.

(3) In the introduction, it would be helpful for the authors to specify exactly what is meant by chaperone. Given the context, it seems that the authors refer to the chaperone activity as one that prevents aggregation. Is this correct?

We refer to chaperone activity specifically as the ability to prevent aggregation of proteins. We have now clarified this definition in the Introduction section of the revised manuscript.

(4) The results for experiments shown in Figure 3 need additional setup in the text. Were these measurements taken immediately after mixing WT, DM, or P33A with polyP? If so, why do condensates immediately appear and then dissipate before ThT-detected aggregates begin forming? Or were condensates allowed to form and then transferred to a different buffer, after which measurements were taken? Without a brief description of the experimental setup, interpreting the results is difficult.

The condensates appear immediately after adding polyP to protein solutions, indicating that the condensate phase is kinetically accessible on mixing polyP with DM or the WT. As illustrated in Figure 3A and 3B, for WT protein, the condensates undergo liquid to solid transition over the time as this likely is the most thermodynamically stable phase. Effectively, this work is to convey that it is important to look at time-dependence of even droplets when formed as they may not be the most stable phase.

(5) Please include images of P33A over the time course of the experiment in Figure 3B.

We have included the representative images of P33A in presence of polyP over the time in Figure 3B in the revised manuscript.

(6) In Figures 3D, E, G, and H, please plot each measurement separately with mean and standard deviation to enable the reader to see each data point.

We have now revised Figures 3D, E, G, and H to show individual data points along with the mean and standard deviation.

(7) In the top paragraph on page 12, "fast-moving molecules" can be replaced with "dynamic molecules", as this offers a better description of the FRAP data.

We have incorporated the suggested changes.

(8) In the "Structural changes within the condensates spans over three hours" results section on page 15, the conclusion reads "In summary, we find that both the WT and the DM 'unfold' on forming condensates with polyP..." The way this is written suggests that WT and DM behave in a similar manner. Given the CD data, however, it seems that by 4 hours, DM forms alpha helices while the WT does not. This suggests that while each unfolds, the conformation at 4 hours is different. The summary should reflect these differences.

We fully agree with the reviewer on this. The summary is now modified to include the fact the DM forms alpha helices at 4 hours while the WT does not.

(9) At the end of the first paragraph of the results section "DNA does not discriminate the conformational ensembles" the authors should refer to Figure 2G, where they show the altered morphology of polP-P33A condensates.

We have now included the reference to Figure 2G.

(10) The authors refer to droplets "solubilizing" throughout the manuscript. It seems that dissolve is a better term to use. Solubilize is better associated with individual biomolecules while dissolve is better associated with condensate behavior.

We thank the reviewer for pointing this out. We have revised the manuscript to replace “solubilize” with “dissolve”.

(11) In Figures 5L and 5N, please change the Y-axis scale so that each curve is visible on the plot.

We have adjusted the Y-axis scale in Figures 5L, 5M, and 5N to ensure that each curve is clearly visible and for easier comparison among the variants.

(12) The authors should show an image of FruR WT and Y19A with DNA for a direct comparison with experiments in which FruR and polyP were used. The addition of turbidity measurements of samples shown in Figure 6D will offer another direct comparison. As written, there is no way for the author to directly compare the effects of polyP and DNA on FruR phase transitions.

As suggested, we have now included representative images of FruR WT and Y19A with DNA (Figure 6K and 6L) to enable a direct comparison with the FruR–polyP experiments. Also, we have already shown turbidity measurements in Figure 6B and 6C corresponding to the samples shown in Figure 6D.

**Reviewer 2:**
In this study, Goyal et al demonstrate that the assembly of proteins with polyphosphate into either condensates or aggregates can reveal information on the initial protein ensemble. They show that, unlike DNA, polyphosphate is able to effectively discriminate against initial protein ensembles with different conformational heterogeneity, structure, and compactness. The authors further show that the protein native ensemble is vital on whether polyphosphate induces phase separation or aggregation, whereas DNA induces a similar outcome regardless of the initial protein ensemble. This work provides a way to improve our mechanistic understanding of how conformational transitions of proteins may regulate or drive LLPS condensate and aggregate assemblies within biological systems.

We thank the reviewer for the favorable comments on the manuscript.

Major Concerns:(1) The authors are using bacterial proteins (CytR and FruR) and solely represent polyphosphates as polyP45 (a polyphosphate with 45 Pi units). However, in bacterial systems, polyphosphates can be significantly longer (in the order of 100s to 1000 Pi units). Additionally, the experiments were run at neutral pH (7.0), and though this is fairly appropriate for the cytoplasm, volutin granules (where polyphosphates often accumulate) are typically considered slightly acidic (pH 5.5-6.5). From a physiological perspective, understanding how pH and the length of polyphosphate influence the ability to induce condensates or aggregates could be of importance.

We appreciate the reviewer’s insightful comments regarding the physiological relevance of polyphosphate length and pH. In our current study, we used polyP45 as it is easily available commercially and we conducted our experiments at pH 7 to mimic the general cytoplasm conditions. We agree that polyphosphates in bacterial cells can be significantly longer (hundreds to thousands of Pi units) and conducting experiments at slightly more acidic environment would be physiologically relevant. We plan to use longer polyP from Regene Tiss Inc and acidic pH to explore how polyphosphate-induced phase separation of CytR vary with pH as a part of a future study. One could imagine doing all the experiments listed in the manuscript at different pH conditions for the different variants, but this could not be a part of the current work which has a specific focus on the differences in maturation properties depending on the nature of starting ensemble. However, the pKa values of the internal hydroxyl groups is ~2.2 (DOI:10.2147/IJN.S389819) indicating that the polyP carries near identical charges in the pH range between 4-7, and hence we expect little change in the charged status of polyP. On the other hand, the protonation states of charged amino acids within CytR could vary with pH, thus influencing its assembly properties.

(2) In the study, the longest metastable condensate induced by polyphosphate lasted approximately 3 hours before resolubilizing. It would be nice if the authors were able to generate a longer-lived condensate phase that would enable further mechanistic studies (e.g., NMR).

We agree that generating longer-lived condensates would be highly valuable for mechanistic studies. However, the formation and stability of condensates is an intrinsic property of protein, and optimizing different conditions for a longer-lived condensate phase is beyond the scope of the current study. It is possible that the condensates are long-lived with longer polyP, but it is not clear if this would indeed be the case. We would also like to state here that while it is common to report on the liquid-to-solid transition in condensates, the intrinsic metastability of droplets (when there is no aggregation) is rarely reported. One possibility is to mutationally introduce cysteine residues and induce the formation of disulphide bridges (as done in a recent work, doi: 10.1021/jacs.4c09557) that make the condensate highly stable kinetically; however, this would also complicate the interpretation as the mechanism of condensate formation might be very different. We have therefore reported our results as an observation arising from differences in the nature of the poly-anionic polymers.

(3) The authors showed that CytR DM (fully folded), CytR WT (minor state folded), and CytR P33A (highly disordered) with polyphosphates lead to longer-lived condensates that resolubilize, shorterlived condensates that aggregate, and immediate aggregating, respectively. Whereas FruR (folded) and FruR Y19A (molten globular) with polyphosphate induce spontaneous aggregation and short-lived condensates, respectively. I would expect FruR to be more similar to CytR DM and FruR Y19A more similar to CytR WT in terms of structure and conformational dynamics and plasticity, yet they have opposing results. This raises a bit of concern. Meaning, that though polyphosphate discriminates between the different ensembles, is it actually possible to obtain information on the initial ensemble composition?

In the current study, we show that CytR WT (less structured) and FruR Y19A (molten globule) form short-lived condensates that aggregate. We agree with the reviewer that while CytR DM (fully folded) forms condensates that dissolve over time, FruR WT (fully folded) variant forms aggregates immediately upon polyP addition. The observations show that polyP can discriminate between different protein conformations, in contrast to DNA, which does not show such selectivity. However, we acknowledge that while polyP-induced behavior reflects aspects of protein ensemble properties, it does not provide direct insight into the nature of the initial conformational ensemble.

(4) In the case of FruR with polyphosphate, no CD for the secondary structure analysis was provided as it was for CytR. It would be useful to see if the polyphosphate-induced structural changes observed for CytR hold true for FruR as well.

We thank the reviewer for the suggestion. In response, we have performed far-UV CD experiments on FruR variants in the presence of polyP. Similar to the CytR WT, FruR WT shows unfolding upon polyP addition. A similar outcome is noted for the Y19A variant though there is significant residual helix content in the condensate unlike the WT. The CD spectra of FruR variants have been added to Figure 6.

Minor Concerns/Suggestions:Under conclusion, third paragraph, first sentence. This sentence reads, "Our observations thus establish that polyP efficiently discriminates the conformational features of proteins than DNA, contributing to the diverse outcomes."

We thank the reviewer for pointing this out. The sentence has been revised for clarity. It now reads “Our observations establish that polyP is more sensitive to the conformational features of proteins than DNA, thereby contributing to the diverse outcomes.”

One experimental suggestion. Seeing that protein dynamics and plasticity seem to play a role. For either CytR WT or DM, it would be interesting to see the influence of temperature. Altering the temperature is a good way to perturb the population distribution of conformation sub-states and to alter kinetics. It may be that at a lower temperature (maybe 5C) for the WT you reduce conformational dynamics and you obtain results more similar to that of the DM. Alternatively, heating the DM would be another option. Obviously, there are additional challenges that may arise with changing the temperature, but if it were to work I think it could add some value.

We thank the reviewer for the thoughtful suggestion. Due to limitations in our current experimental setup (as the reviewer notes as ‘challenges’)- the confocal set up does not have a temperature controller - we will not be to perform temperature-controlled assays. However, the ‘structure’ of CytR variants do not vary much between 280 – 298 K, and this is one of the reasons for choosing three variants without altering any other thermodynamic property. If temperature were varied, the dynamics of polyP would also change and hence the true molecule origins of any differences we might observe will be confounded by the dynamic effects on polyP as well. In this work, we have eliminated any dynamic differences in polyP by performing the experiments at a fixed temperature.